# Room-temperature direct synthesis of semi-conductive PbS nanocrystal inks for optoelectronic applications

Yongjie Wang[1,2,5], Zeke Liu[1,5], Nengjie Huo[2], Fei Li[1], Mengfan Gu[1], Xufeng Ling [1], Yannan Zhang[1], Kunyuan Lu[1], Lu Han[1], Honghua Fang[3], Artem G. Shulga[3], Ye Xue[1], Sijie Zhou[1], Fan Yang[1], Xun Tang[1], Jiawei Zheng[1], Maria Antonietta Loi[3], Gerasimos Konstantatos[2,4] & Wanli Ma [1]*

Lead sulphide (PbS) nanocrystals (NCs) are promising materials for low-cost, high-performance optoelectronic devices. So far, PbS NCs have to be first synthesized with long-alkyl chain organic surface ligands and then be ligand-exchanged with shorter ligands (two-steps) to enable charge transport. However, the initial synthesis of insulated PbS NCs show no necessity and the ligand-exchange process is tedious and extravagant. Herein, we have developed a direct one-step, scalable synthetic method for iodide capped PbS (PbS-I) NC inks. The estimated cost for PbS-I NC inks is decreased to less than 6 $·g$^{-1}$, compared with 16 $·g$^{-1}$ for conventional methods. Furthermore, based on these PbS-I NCs, photo-detector devices show a high detectivity of $1.4 \times 10^{11}$ *Jones* and solar cells show an air-stable power conversion efficiency (PCE) up to 10%. This scalable and low-cost direct preparation of high-quality PbS-I NC inks may pave a path for the future commercialization of NC based optoelectronics.

[1] Jiangsu Key Laboratory for Carbon-Based Functional Materials &Devices, Institute of Functional Nano & Soft Materials (FUNSOM), Joint International Research Laboratory of Carbon-Based Functional Materials and Devices, Soochow University, 199 Ren'ai Road, Suzhou 215123, People's Republic of China. [2] ICFO-Insitut de Ciencies Fotoniques, The Barcelona Institute of Science and Technology, Castelldefels, 08860 Barcelona, Spain. [3] Zernike Institute for Advanced Materials, University of Groningen, Nijenborgh 4, 9747 AG Groningen, Netherlands. [4] ICREA-Institució Catalana de Recerca i Estudia Avançats, Lluis Companys 23, 08010 Barcelona, Spain. [5] These authors contributed equally: Yongjie Wang, Zeke Liu. *email: wlma@suda.edu.cn

L ead chalcogenide nanocrystals (NCs) are promising materials with high bandgap-tunability for flexible and lightweight infrared optoelectronics[1,2]. Devices based on lead chalcogenide NCs can be processed using industrially friendly solvents under room temperature, which is compatible with low-cost, large-scale manufacturing on plastic substrates[3–6].

Lead sulfide (PbS) NCs can be synthesized using a variety of reported methods[7–10]. Although many different methods have led to high-efficiency devices, the most common approach is the lead oxide (PbO) and hexamethyldisilathiane (TMS-S) hot injection route pioneered by Hines and Scholes[9]. The NCs synthesized in these routes are generally accomplished by the use of precursors with long-alkyl chains surfactants, which served as surface ligands for the NCs. Although excellent control over the optical properties and morphologies of these NCs can be achieved through the organic surfactants[11], these long chain ligands hamper the electronic transport between NCs, leading to poor device performance[12,13]. One sophisticated approach for solving this issue has been through post-synthetic ligand exchange method whereby these long organic ligands can be replaced by shorter organic molecules or single atoms/ions[14–16].

A range of ligand exchange strategies have been explored to reduce the inter-particle spacing for enhanced carrier transport and, at the same time, achieve excellent surface passivation to reduce trap induced recombination losses[17–19]. Conventionally, ligand exchange is carried out in solid-state using time-consuming layer-by-layer (LbL) process to build up thick device-level films[15,20,21]. Recently reported solution-phase ligand exchange can provide NCs inks that can be deposited in a single step[16,22,23]. Efficient PbS NCs devices have been achieved using this method with lead halides or organic molecules as the exchange ligands[23,24]. However, the initial synthesis of long organic-ligands capped PbS NCs show no necessity for the final film deposition and the ligand exchange process is still tedious and extravagant[25]. The cost of present PbS NC inks is still estimated higher than 16 $ g$^{-1}$, which hinders the commercial possibility of PbS NC solar cells[25]. Furthermore, large-scale NC inks preparation is also critical for high-throughput manufacturing of NCs based optoelectronics[26].

Herein, we have developed a new method to direct synthesize halide capped semi-conducting metal chalcogenide NC inks for optoelectronic applications. Metal halide and N, N-Diphenyl thiourea (DPhTA) are employed as precursors. Based on this method, in situ halide capped PbS, silver sulfide (Ag$_2$S) and cadmium sulfide (CdS) NC inks are directly synthesized at room temperature, showing excellent scalability and versatility. We have also demonstrated a lab scale-up synthesis to increase the reaction volume to 2 L and obtained more than 88 g PbS NC ink solids in one pot, with the cost of PbS NC inks lower than 6 $ g$^{-1}$. Furthermore, Obtained PbS NC inks show good size tunability and state-of-the-art performance for optoelectronic applications. Based on these PbS NC inks, photodetectors show a high detectivity up to $1.4 \times 10^{11}$ Jones at 635 nm with a fast response time less than 4 ms and photovoltaic devices show a power conversion efficiency (PCE) up to 10% with superior device air stability, which outperform the control devices based on solution-phase ligand exchanged NCs.

## Results

### Direct synthesis and characterization of iodide capped PbS NC inks

Conventionally, in order to obtain semi-conductive films, oleic acid capped PbS (PbS-OA) NCs have to be first synthesized by hot injection method and then ligand exchanged with shorter organic ligands or inorganic halide ligands to improve conductivity by either LbL solid-state ligand exchange or solution-phase ligand exchange method, which is tedious and time-& cost-consuming[25], as shown in Fig. 1a. Here, we have developed a direct and low-cost synthetic method for iodide capped PbS (PbS-I) NC inks. The synthesis of the PbS-I NC inks, as depicted in Fig. 1b, is carried out by a simple and fast one-step injection at room temperature under nitrogen. We use dimethyl formamide (DMF) to dissolve lead iodide (PbI$_2$) and diphenyl thiourea (DPhTA) with no degassing process that is usually required for

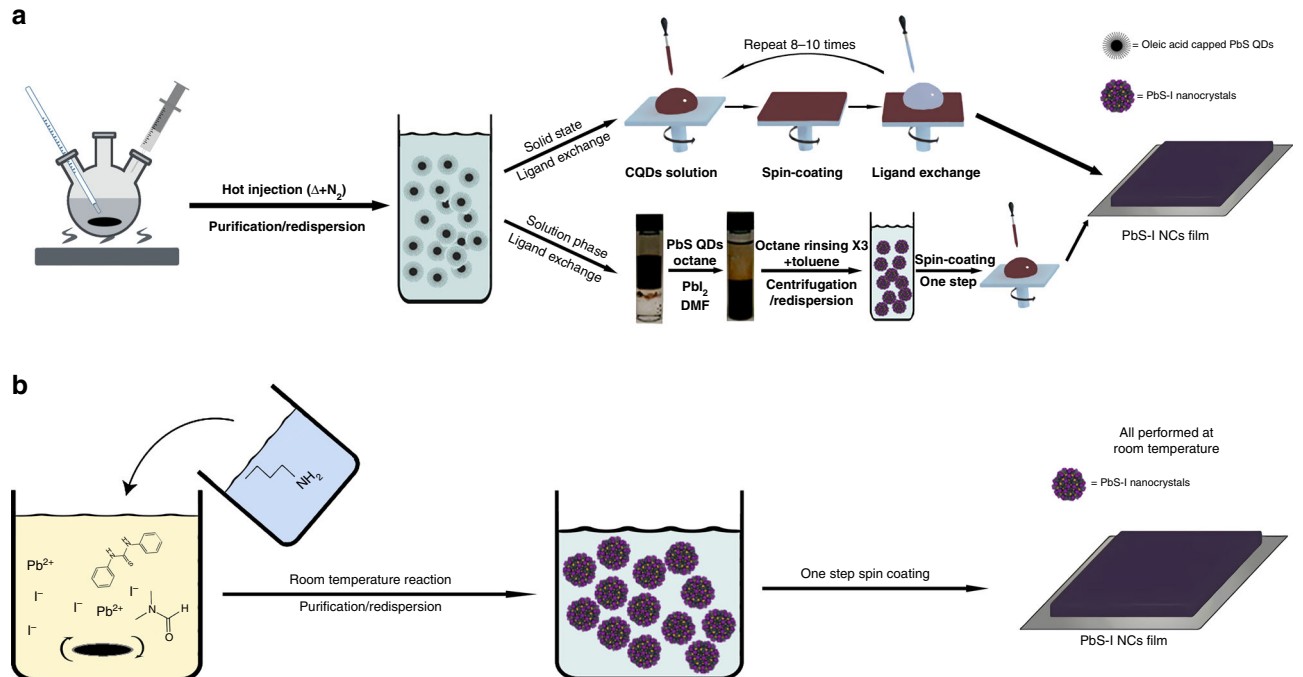

**Fig. 1** Schematic representation of different preparations for PbS-I NC films. **a** Conventional two-step "synthesis to ligand-exchange" process and **b** our one-step direct synthesis of iodide capped PbS NCs inks

conventional PbS synthesis. After butyl amine (BA) was injected into the aforementioned solution, the solution turned from light yellow to black immediately, indicating the formation of PbS NCs. Since $PbI_2$ and DPhTA cannot react in the absence of BA, we speculate that the $^-SH$ converted from DPhTA under alkaline condition works as the real sulfur precursor (Supplementary Fig. 1)[27,28]. We also tried to replace DPhTA with TMS-S, which can react with $PbI_2$ directly. The reaction could indeed be triggered without BA and, however, resulted in large size PbS precipitates (Supplementary Fig. 2). If BA was introduced, the obtained PbS NCs showed almost similar size and dispersibility with our $PbI_2 +$ DPhTA + BA recipe, which indicates BA also function as temporal surface ligands to control the nucleation and growth in addition to promote the release of $^-SH$ from DPhTA. The detailed discussion of reaction mechanism was presented in the Supplementary Note 1.

The PbS-I NCs can be separated by anti-solvent centrifugation for one time and the solids can be re-dispersed in DMF with concentration up to $800\,mg\,mL^{-1}$, after which the inks can be used directly for device preparation. The size of the PbS-I NCs can be easily controlled by varying the $PbI_2$ to DPhTA ratio (Pb/S ratio). With the decrease of Pb/S ratio, larger NCs can be obtained, which is evidenced from the redshift of both absorption and photoluminescence (PL) peaks as shown in Fig. 2a, b. Since the synthesis process was performed at room temperature (Supplementary Figs. 3, 4 and 5), it could be easily scaled-up to a litre-sized synthesis (2000 mL, Supplementary Fig. 6, 7). It is worth noting that DMF is not an optimal solvent for commercialization. Thus, we chose a more environmentally benign solvent γ-butyrolactone (GBL)[29] for the synthesis of PbS-I NCs. As shown in Supplementary Figs. 8 and 9, we have successfully synthesized PbS-I NCs using non-toxic GBL as the reaction solvent. In addition, devices based on GBL synthesized PbS-I NCs show comparably high performance as well (Supplementary Fig. 10).

Important difference between our one-step reaction scheme and the previous reported "synthesis to ligand-exchange" (two-step) methods is that we use scalable room temperature reactions, and NCs are in situ passivated during synthesis by halide ions. As solution phase exchanged PbS NC inks, these NCs can be directly deposited to form thick and uniform films in only one single step. In addition, the sulfur precursor used here is diphenylthiourea (DPhTA), which is more environment friendly and cheaper than conventionally used TMS-S[9,30]. Therefore, our method is more amenable to be scaled-up in industrial processes by saving the cost of both processing and materials (Supplementary Note 2, Supplementary Tables 1 and 2).

As shown in Fig. 2c, d, based on the images of transmission electron microscope (TEM), the crystalline PbS-I NCs have an average size of $3.46 \pm 0.58$ nm. The lattice of the NCs matches well with cubic phase PbS structure, as shown in Fig. 2e. According to X-ray photoelectron spectroscopy (XPS) analysis, as shown in Fig. 3a, b, the iodide to lead (I/Pb) ratio is 0.84 in a spin-coated film of PbS-I NCs, which is comparable to the highest reported halide/lead ratio((I + Br)/Pb = 0.86) in solution-phase ligand exchange method[16], indicating an exceptional passivated surface (Supplementary Fig. 11). Pb-OH groups bound on the facet of PbS NCs are considered as one of the main sources for sub-bandgap states[31]. From the O 1s XPS spectrum shown in Fig. 3c, there is a small amount of oxidized butyl amine on the surface of NCs (Supplementary Fig. 12) and the Pb-OH/Pb ratio is only 0.04, which indicates a low trap states density in these PbS-I NCs[31].

Furthermore, considering $Ag_2S$ and CdS are also commonly used materials in photovoltaic devices[32,33], we further explored

the synthesis of halide capped $Ag_2S$ and CdS NCs. With some modifications (See the "Methods" section), Accanthite phase $Ag_2S$ NCs and Hawleyite phase CdS NCs were successfully synthesized (Supplementary Figs. 13 and 14) with $I^-$ and $Cl^-$ ions as the capping ligands, respectively. These results indicate the versatility of this synthetic method and its potential to become a general synthetic routine for obtaining other metal chalcogenide NCs inks.

**Optoelectronic characterization and device performance.** For exploring the basic electronic properties of PbS-I NC films, carrier density is calculated to be $8.77 \times 10^{16}\,cm^{-3}$ by capacitance-voltage test (Supplementary Note 3, Supplementary Fig. 15), which is also similar with that of ligand exchanged PbS NCs[34,35]. In order to further explore the surface passivation and trap states of PbS-I NCs, time-resolved PL has been tested. As shown in Fig. 3d, PbS-I NCs in DMF solution show a very long average PL lifetime of 1.90 μs, which is even close to that of oleic acid capped PbS QDs[36], indicating a good surface passivation of these PbS-I NCs. The trap density of PbS-I NCs film is then measured by the space charge limited current (SCLC) method[35]. As shown in Fig. 3e, the trap density can be calculated as $7 \times 10^{15}\,cm^{-3}$ for PbS-I NCs film (Supplementary Note 4), which is even lower than the reported $1.6 \times 10^{16}\,cm^{-3}$ for solution-phase ligand exchanged PbS NCs films[37]. Furthermore, field-effect transistors (FETs) measurements were carried out to study the mobility of PbS-I NCs films. The FETs were fabricated on a highly doped silicon substrate covered with a 230 nm thick $SiO_2$ as gate dielectric layer and photo-lithographically patterned gold electrodes as source and drain. Figure 3f shows the transfer characteristics at a drain voltage of 15 V. The carrier mobility is calculated from the slope of the $I_{DS}$ versus $V_{GS}$ plot in the linear regime according to the equation[38],

$$\mu = \frac{L}{WV_{DS}C_i} \cdot \frac{\partial I_{DS}}{\partial V_{GS}} \quad (1)$$

Where $\mu$ is the carrier mobility, $I_{DS}$ is the drain current, $V_{GS}$ is the gate voltage, $L$ and $W$ are the channel length (10 μm) and channel width (10 mm), respectively; $C_i$ is the gate capacitance per unit area. The electron mobility of PbS-I NC film is calculated as $0.0267\,cm^2\,V^{-1}\,s^{-1}$, which is close to the reported mobility of solution-phase ligand exchanged PbS NCs[37,39].

For further practical demonstration of the synthesized PbS-I NC inks, broadband photodetectors were fabricated and characterized. Figure 4a shows the schematic structure of the photodetector with a channel length and width of 20 μm and 1 mm, respectively. For a photodetector, responsivity ($R$) and specific detectivity ($D^*$) are two key figures-of-merit[40]. $R$ is defined as $R = \frac{I_{ph}}{P_{in}}$, representing the ratio of photocurrent generated in the detector over the incident optical power in units of $A \cdot W^{-1}$, where $I_{ph}$ is the photocurrent and $P_{in}$ is the incident light power on the effective area[41]. Figure 4b shows the photocurrent and responsivity of the PbS-I NCs based photodetectors as a function of light power density, exhibiting a high $R$ of $1.5\,A\,W^{-1}$ at low irradiance, which is evidently higher than solution-phase ligand exchanged devices ($0.56\,A\,W^{-1}$, Supplementary Fig. 16). $D^*$ is commonly used to characterize the sensitivity of a detector, which can enable the comparison of detectors with different geometries. The unit is $cm \cdot Hz^{1/2} \cdot W^{-1}$ or $Jones$. The specific $D^*$ can be defined as following equation:[42]

$$D^* = \frac{\sqrt{AB}}{NEP} = \frac{R\sqrt{A}}{S_n} \quad (2)$$

where NEP is the noise equivalent power in W, $R$ is the responsivity in $A \cdot W^{-1}$, $A$ is the active area of the detector in

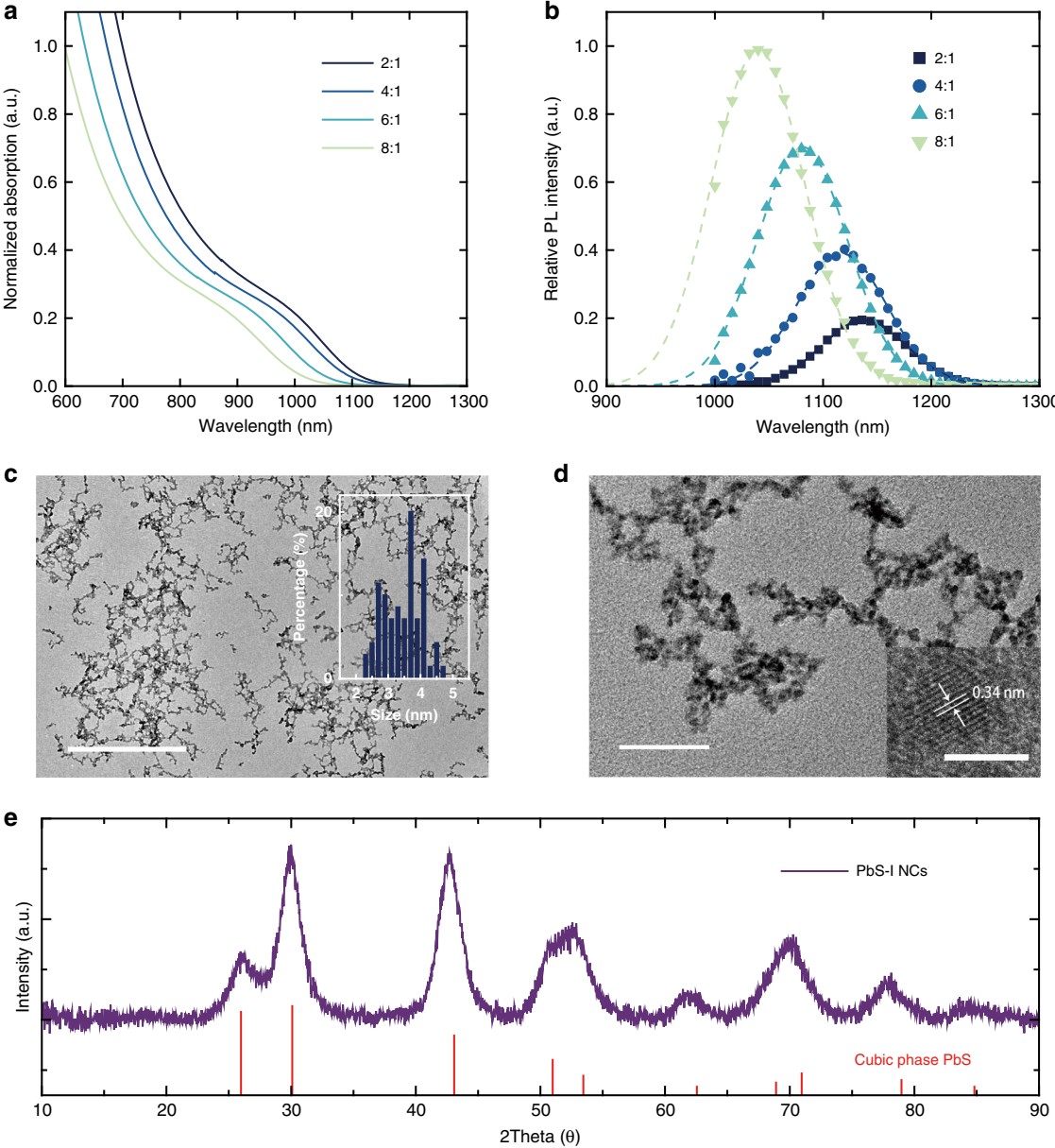

**Fig. 2** Characterization of direct-synthesized PbS-I NCs inks. Absorption (**a**) and PL spectrum (**b**) of synthesized PbS-I NCs with different Pb/S Precursor ratio. Transmission Electron Microscopy (TEM) images (**c**, **d**) and X-ray diffraction (XRD) pattern (**e**) of PbS-I NCs, indicating a cubic phase PbS crystal lattice. (Scale bar: 500 nm in (**c**), 50 nm in (**d**) and 5 nm in (**d**) inset.)

cm$^2$, $B$ is the noise bandwidth in Hz, and $S_n$ is the noise current spectral density of the detector in A·Hz$^{-1/2}$. For the NEP of detectors, the time-resolved dark current is measured with applying a constant drain bias. By taking the Fourier transform of dark current traces, the noise power densities are obtained as shown in Supplementary Fig. 17, and then the noise spectral densities $S_n$ are extracted at frequency of 1 Hz, under the same conditions as the optical measurements were performed, resulting in noise values of 0.15 pA Hz$^{-1/2}$. The spectral $D^*$ and $R$ for PbS-I NCs based devices has been plotted in Fig. 4c, which shows a broad spectral response up to 1.1 μm and is consistent with the absorption spectral of PbS-I NCs. The $D^*$ at 635 nm light is $1.4 \times 10^{11}$ *Jones*, which is an order of magnitude higher than solution-phase ligand exchanged devices (Supplementary Table 3). Furthermore, if only shot noise is taken into account, the shot noise estimated detectivity can be calculated to be $1.78 \times 10^{13}$ *Jones*, which is, based on the same calculation method, superior than the

performance of recently reported PbS QDs based bilayer structure[41,43]. In addition to the excellent $R$ and $D^*$, our detectors also exhibit a high light-to-dark current ratio up to around $5 \times 10^3$ and fast response speed (Fig. 4d). From the time traces of photocurrent with light irradiation on/off in Fig. 4d, the devices show a well photocurrent on/off repeatability and fast speed with rise and decay time of less than 4 ms, which is limited by the measurement system resolution. These excellent photodetection performances are benefited from the high optical and electronic quality of the PbS-I ink, which may be due to the low density of electronic defects and reasonably high mobility in the NCs array film.

In addition to photodetectors, iodide-passivated PbS NCs are also promising absorbers in high-performance photovoltaics. In order to investigate these NCs inks in solar-cell devices, we have selected a widely used architecture:[15,16] indium doped tin oxide coated glass as the substrate, a layer of solution processed zinc

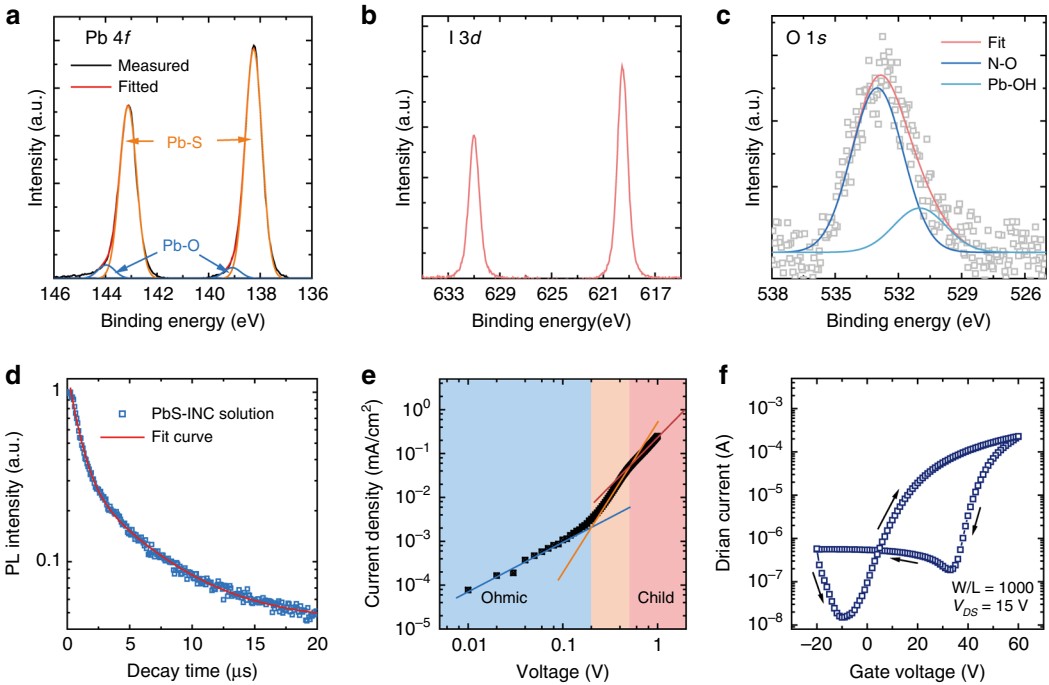

**Fig. 3** X-ray Photoelectron Spectrum (XPS) and optoelectronic properties. **a** Pb 4*f* XPS spectrum, **b** I 3*d* XPS spectrum and **c** O 1*s* XPS spectrum of PbS-I NCs. **d** Transient photoluminescence of PbS-I NC solution. **e** *J-V* curve of space charge limited current device of PbS-I NCs. Device structure: Ag/PbS-I/Ag. **f** Transfer characteristics of PbS-I NCs based FET

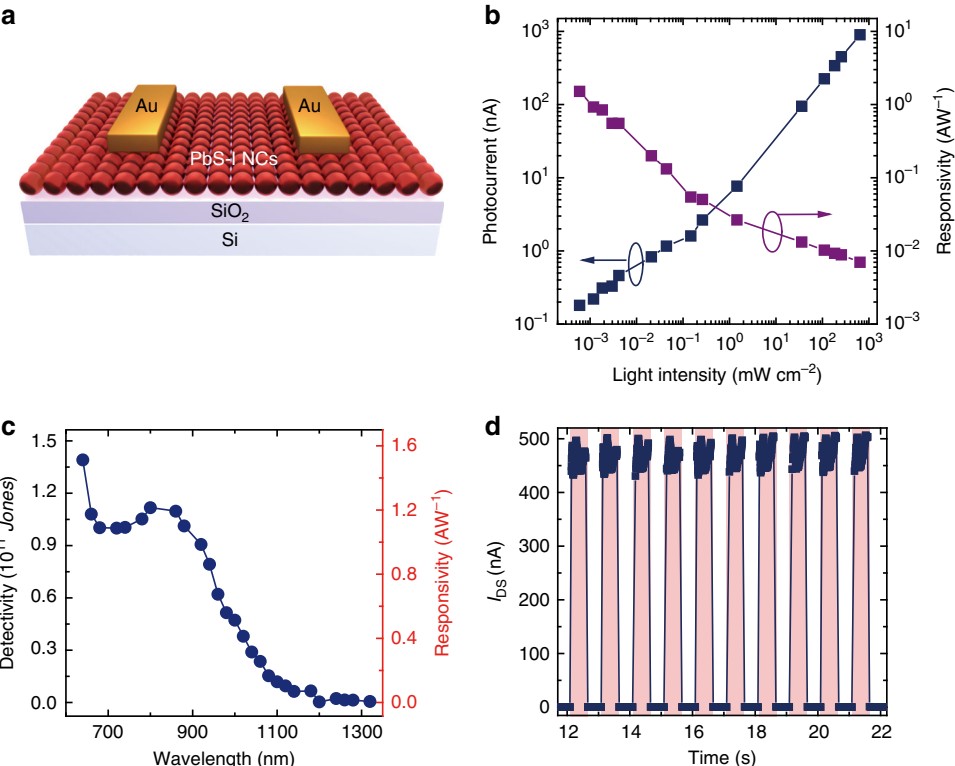

**Fig. 4** Photodetectors based on PbS-I NCs. **a** Scheme of the structure of photodetector devices. **b** Photocurrent and responsivity under different light intensity of 635 nm laser. **c** Spectral detectivity and responsivity of PbS-I NCs based photodetectors. **d** Dynamic response of the device upon on-off switching of 635 nm laser. The measurements are carried out at a drain bias of 5 V and without gate bias

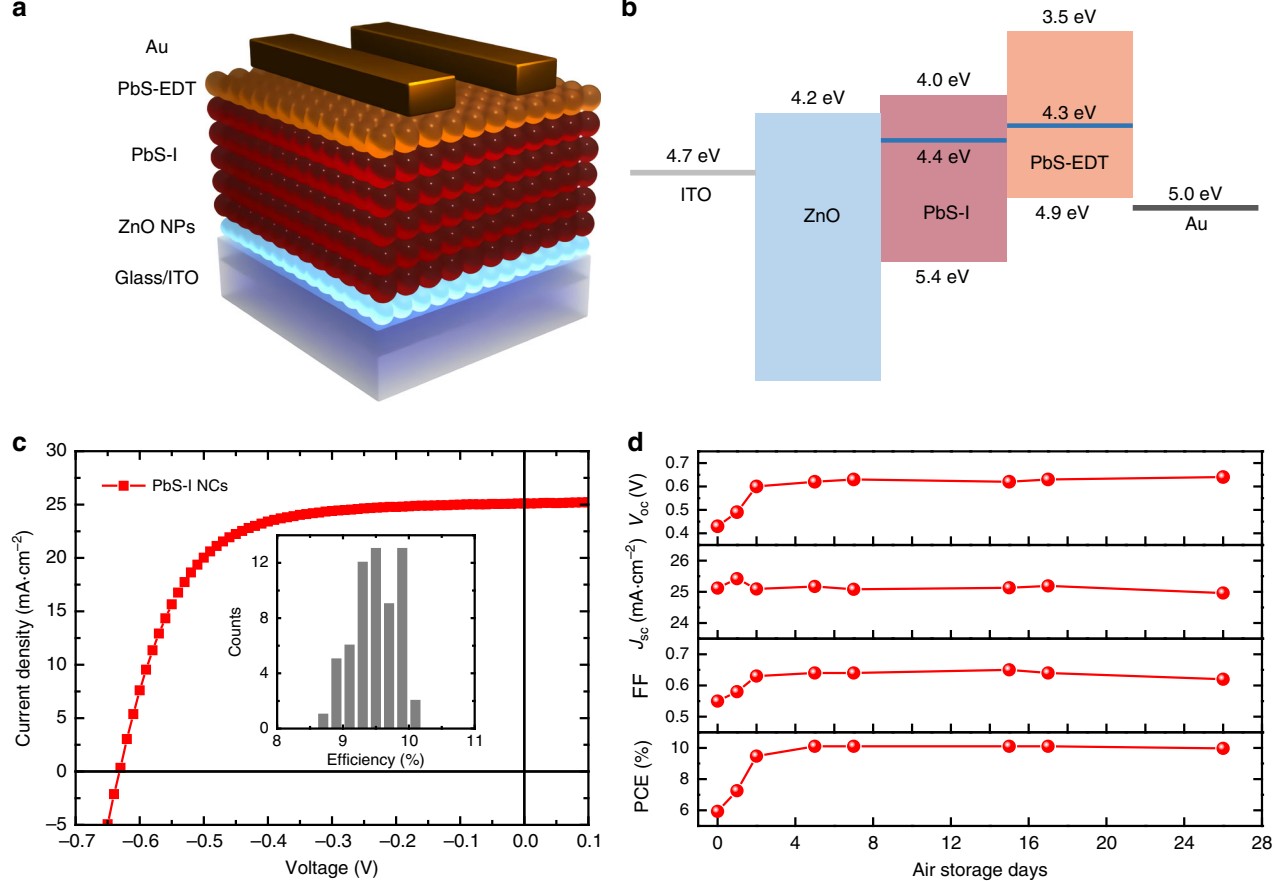

**Fig. 5** Photovoltaic devices based on PbS-I NCs. **a** Scheme of the device architecture of the solar cell. **b** Energy diagram of the photovoltaic device. **c** *J-V* curve of optimized PbS-I NCs device. Inset is histogram of device efficiency based on 61 devices. **d** Air stability of PbS-I NCs based photovoltaic devices

oxide (ZnO) nanoparticles as the electron-extracting layer, one layer of spin-coated PbS-I NCs as the photoactive layer, two layers of ethanedithiol (EDT) treated PbS-OA NCs as the hole-transporting layer and evaporated gold (Au) as the top contact. In Fig. 5a we scheme the device architecture and in Fig. 5b we present the energy levels of each layer. For the electrodes and the charge-extracting layers, we have used values reported in the literature[15,44], while for the PbS-I NCs layer, the energy level of the conduction and valence band edges have been extracted by adding the optical bandgap to the position of the valence band maximum as determined by ultraviolet photoelectron spectroscopy (UPS) measurements (Supplementary Fig. 18).

With a single spin-coating of the suspended PbS-I NC inks onto the ZnO substrate, a device-level thick film could be obtained. After 5 days' air storage, photovoltaic devices based on PbS-I NCs show an average PCE of 9.48% and the champion device exhibits a short-circuit current density ($J_{sc}$) of 25.1 mA cm$^{-2}$, an open-circuit voltage ($V_{oc}$) of 0.63 V, and a fill factor (FF) of 0.64, leading to a power-conversion efficiency (PCE) of 10.14%, as shown in Fig.5c, which is slightly higher than that for solution-phase ligand exchanged devices in our lab (Table 1, Supplementary Figs. 19 and 20). The high $V_{oc}$ and FF confirm the excellent passivation of the as-synthesized PbS-I NCs. Meanwhile, as shown in Supplementary Fig. 21, the $J_{sc}$ is also calculated from the external quantum efficiency (EQE) integration, which is in good agreement with the results from the *J-V* measurement. Furthermore, photovoltaic devices fabricated with PbS-I NC inks synthesized in large-scale show a decent PCE up to 9% (Supplementary Fig. 22), which indicates the good scalability of this room temperature direct synthesis method. Considering the device stability is closely related

**Table 1 Photovoltaic parameters of devices based on different PbS NCs inks**

|  | $V_{oc}$ (V) | $J_{sc}$ (mA cm$^{-2}$) | FF | PCE (%) |
|---|---|---|---|---|
| Direct synthesis | | | | |
| Average | 0.61 ± 0.01 | 25.1 ± 0.3 | 0.62 ± 0.01 | 9.48 ± 0.35 |
| Champion | 0.63 | 25.1 | 0.64 | 10.1 |
| Solution-phase ligand-exchange | | | | |
| Average | 0.58 ± 0.02 | 24.4 ± 0.6 | 0.64 ± 0.02 | 9.11 ± 0.38 |
| Champion | 0.61 | 24.6 | 0.66 | 9.95 |

Note: Statistics are based on 61 different devices

with the surface passivation of employed NCs[31,45], air stability of PbS-I NCs based solar cells has been tested. After storage in ambient air under an average humidity of 30% for one month without encapsulation, the solar cells can still maintain more than 98% of the original efficiency (Fig. 5d), which further confirms the high quality of in situ passivation of PbS-I inks. In addition, PbS-I NCs synthesized from concentrate solution showed higher performance, as shown in Supplementary Fig. 23. With purer lead iodide and more concentrate precursor solution, devices may provide even higher performance.

## Discussion

In summary, we have developed a direct one-step and scalable synthetic method for halide capped semi-conducting metal chalcogenide (PbS, Ag$_2$S, and CdS) NC inks. This method enables

the in situ iodide passivation of PbS NCs and the fabrication of thick films without ligand exchange. Due to the reduced processing-steps, and the use of cheaper precursors, the estimated cost is reduced from more than 16 $ g$^{-1}$ for previously reported PbS inks to lower than 6 $ g$^{-1}$ for our directly synthesized PbS-I inks. We have also easily demonstrated a lab scale-up synthesis, benefiting from the room temperature reaction. In addition, the obtained PbS NC inks show state-of-the-art performance for optoelectronic applications. Based on these PbS-I NCs, photodetector devices show a high detectivity of $1.4 \times 10^{11}$ Jones with a fast response time less than 4 ms and solar cells show an air-stable PCE more than 10%, which indicates the excellent in situ passivation of the NCs ink during synthesis. Furthermore, this simple, scalable and low-cost preparation of high-quality PbS NCs may pave a path for the future commercialization of NC based optoelectronics.

## Methods

**Chemicals**. Lead iodide (PbI$_2$, 99+ %), silver iodide (AgI, 99.9%), cadmium chloride (CdCl$_2$, 99.99%), N,N′-diphenyl thiourea (DPhTA, 98%), lead acetate trihydrate (PbAc$_2$·3H$_2$O, 99%), hexamethyldisilathiane (TMS-S, 98%), ammonium acetate (AA, >98%), ammonium chloride (NH$_4$Cl, 98%), oleic acid (OA, 90%), 1-octadecene (ODE, 90%), hexane (95%), isopropanol (IPA, 95%), acetone (95%), toluene (95%), N,N′-dimethylformamide (DMF, 99.8% anhydrous), N-methylfomamide (NMF, 99.8%), butyl amine (BA, >99%), 1,2-ethanedithiol (EDT, 98%), acetonitrile (ACN, 99%).

**Synthesis and purification of PbS NCs ink**. For oleate capped PbS NCs, the synthesis is the same as reported in our previous work[36]. For the synthesis of PbS-I NCs, 8 mmol PbI$_2$ and 2 mmol DPhTA was dissolved in 9 mL DMF with stirring at room temperature in a nitrogen filled glovebox, after solids all dissolved in DMF, 1 mL butylamine was injected. The solution immediately turned black. After adding toluene as anti-solvent, the PbS NCs were centrifuged for 5 min at 8000 rpm. The solids were stored in glovebox. The solubility of resulted PbS-I NCs could be improved if the synthesis process is proceeded inside nitrogen atmosphere. PbS-I NCs solution was prepared by re-dispersing NC solids in DMF with a concentration of 500 mg mL$^{-1}$ for further devices fabrication. For the PbS-I NCs synthesized with GBL as solvent, 4 mmol PbI$_2$, 4 mmol MAI and 1 mmol DPhTA was dissolved in 9 mL GBL with stirring at room temperature in a nitrogen filled glovebox, after solids all dissolved, 2 mL butylamine was injected. After 5 min reaction at room temperature, toluene was added as anti-solvent and after centrifugation, supernatant was discarded. Solid is dissolved in GBL for further use.

For solution-phase ligand exchanged PbS NCs, 5 mmol PbI$_2$ and 2 mmol NH$_4$Ac (AA) are dissolved in 50 mL DMF. Then, 50 ml 10 mg mL$^{-1}$ PbS-OA NCs in hexane was dropwise added into PbI$_2$/AA/DMF solution. After the PbS NCs completely transferred to the DMF phase, the DMF solution was washed four times with 50 mL hexane. After ligand exchange, NCs were precipitated by adding toluene as anti-solvent and centrifuged at 8000 rpm for 5 min. The PbS NCs were vacuumized for 30 min to yield NCs powder. The dried iodide-passivated PbS NCs powder was then redispersed in butylamine (~200 mg mL$^{-1}$) for device fabrication.

**Synthesis of Ag$_2$S-I NCs and CdS-Cl NCs ink**. For the synthesis of Ag$_2$S-I NCs, 19 mg DPhTA was dissolved in 1 mL DMF and 78 mg AgI was dissolved in 400 μL BA. AgI/BA precursor was swiftly injected into DPhTA/DMF precursor under stirring at room temperature. After 10 min reaction, 4 mL toluene was added and solution was centrifuged for 5 min at 6000 rpm. Supernatant was discarded and solid is dissolved in DMF for further characterization.

For the synthesized of CdS-Cl NCs, 10.7 mg NH$_4$Cl and 36.6 mg CdCl$_2$ were dissolved in 2 mL NMF. 91.2 mg DPhTA (or 12.8 mg sulfur powder) was dissolved into 400 uL BA. Sulfur/BA precursor was rapidly injected into CdCl$_2$/NMF precursor under stirring at room temperature. After 10 min reaction, crude solution was centrifuged directly for 5 min at 6000 rpm. Solid is dissolved in pyridine for further characterization.

**Synthesis of ZnO nanoparticles**. ZnO nanoparticles were synthesized according to the literature with some modification[46]. Zinc acetate dehydrate (2.95 g) was dissolved in 125 mL of methanol at 60 °C. Potassium hydroxide (1.48 g) was dissolved in 65 mL methanol. The potassium hydroxide solution was slowly added to the zinc acetate solution and the solution was kept stirring at 63 °C for 3 h. ZnO nanocrystals were extracted by centrifugation and then washed twice by methanol followed by centrifugation. Finally, 10 mL of chloroform and 10 mL of methanol were added to the precipitates and the solution was filtered with a 0.45 μm filter.

**Solar cell fabrication**. ITO-coated glass substrates were sonicated in sequence with acetone, detergent water, acetone, 2-propanol and acetone again for 15 min, respectively. The substrates were treated with oxygen plasma for 10 min to remove the last traces of organic residues before spin coating. The ZnO NP layers were fabricated by spin-coating the nanoparticle solution onto the ITO substrates. The PbS-I NCs solution was spin coated at 2000 rpm for 45 s, followed by 10 min annealing at 70 °C. The PbS-I NCs layer was proceeded in inert atmosphere to achieve high-performance devices. For the EDT treated PbS NCs layers, 20 mg mL$^{-1}$ oleate capped PbS NCs were spin-coated at 2500 rpm for 20 s, followed with EDT/acetonitrile solution treatment for 30 s and acetonitrile rinsing for twice. Two layers of PbS-EDT were deposited. After 12 h of exposure to dry air, a 100 nm thick gold film was thermally evaporated through a shadow mask to prepare devices with a total active area of about 0.0725 cm$^2$. Devices are stored in ambient air (humidity around 30%) for the air stability test.

**Photodetector fabrication**. Before the device fabrication, the silicon substrate was cleaned by acetone and isopropanol. Metal contacts were then fabricated by the laser writing lithography, and Au (50 nm) electrodes were evaporated by thermal evaporation. Then the PbS-I NCs were spin-coated and annealed the same as solar cell devices.

**Measurement and characterization**. Current-voltage characteristics were recorded using a Keithley 2400 (I-V) digital source meter under a simulated AM 1.5 G solar irradiation at 100 mW cm$^{-2}$ (Newport, Class AAA solar simulator, 94023A-U). The light intensity is calibrated by a certified Oriel Reference Cell (91150 V) and verified with a National Renewable Energy Laboratory calibrated Hamamatsu S1787-04 diode. Voltage swept from −1 to 0.1 V (revrese sweep) and 0.1 to −1 V (forward sweep) with the speed of 0.01 V per point and a dwell time of 10 ms. Devices were tested in ambient air. The electrical and photocurrent measurements were performed in ambient conditions using an Agilent B1500A semiconducting device analyzer. For spectral photo-response measurements, the devices were illuminated with fiber-coupled light from a supercontinuum light source (Super-KExtreme EXW-4, NKT Photonics). Responsivity and temporal response times were measured under short-pulsed light at a wavelength of 635 nm from a four-channel laser controlled with an Agilent A33220A waveform generator. UV-vis-NIR spectra were recorded on a Perkin Elmer model Lambda 750. PL measurements were performed on samples dissolved in DMF. The second harmonic (400 nm) of a Ti:sapphire laser (Coherent, Mira 900, repetition rate 76 MHz) was used to excite the samples. The optical emission was recorded by a cooled array detector (Andor, iDus 1.7 μm). Time-resolved PL spectra were detected using a Hamamatsu streak camera with a cathode sensitive to near-IR radiation. The PbS-I NC films for UPS/XPS test were made by spin-coating 50–100 nm thick film onto glass/ITO substrate.

**Reporting summary**. Further information on research design is available in the Nature Research Reporting Summary linked to this article.

## Data availability

The data that support the findings of this study are available from the corresponding author upon reasonable request.

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

## Acknowledgements

This work was supported by the National Key Research Projects (Grant No. 2016YFA0202402), the Natural Science Foundation of Jiangsu Province of China (BK20170337), the National Natural Science Foundation of China (Grant No. 61674111), and "111" projects. The author thanks the Collaborative Innovation Center of Suzhou Nano Science and Technology, Soochow University. We also acknowledge the Priority Academic Program Development of Jiangsu Higher Education Institutions (PAPD). G.K. acknowledges financial support from Fundacio Privada Cellex, the program CERCA and from the Spanish Ministry of Economy and Competitiveness, through the "Severo Ochoa" Programme for Centres of Excellence in R&D (SEV-2015-0522). This project has received funding from the European Union's Horizon 2020 research and innovation programme under the Marie Skłodowska-Curie grant agreement No. 754558.

## Author contributions

W.M. supervised the study. Y.W. conceived the idea and carried out the synthesis, characterizations of PbS NCs. Y.W., K.L., and Z.L. carried out the TEM test. Y.W. and F.L. fabricated and characterized the solar cell devices. Y.W., N.H., and G.K. fabricated and characterized the photodetector devices. M.A.L., H.F., and A.G.S. carried out the FET fabrication and characterization. Y.W., X.L., Y.Z., carried out the UPS and XPS test. Y.X., M.G., S.Z., F.Y., X.T., L.H., and J.Z. synthesized the ZnO NPs and prepared the solution. The manuscript was mainly written by Y.W. and revised by W.M. All authors discussed the results and provided input to the manuscript.

## Competing interests

The authors declare no competing interests.
