## [Peer Review File · Nature Communications]

Reviewers' comments:

Reviewer #1 (Remarks to the Author):

The manuscript "Room-temperature direct synthesis of semi-conductive PbS nanocrystal inks for optoelectronic applications" is an interesting piece of work for the effective upscaling of PbS quantum dot (QD) fabrication with a significant reduction of the production cost and no significant reduction of performance (even an increase). The authors can synthesize PbS QDs at room temperature and prepare inks able to produce photodetectors and solar cells with performances at level of the state-of-the-art. Taking into account these points I consider that the manuscript could be interesting for the readers of Nature Communications. Nevertheless, I would suggest the revision of some parts:

- Future commercial applications of PbS QDs could be restricted by the use of Pb. The synthesis route reported here is appealing but the use of DMF, that also awake important concerns, likely will not help in the commercialization. Did the authors used another solvents? This discussion is mandatory if the final scope is the commercialization.
- PL measurements (Fig. 2b y S3b) shows a clear queue or shoulder at long wavelengths, is it originated by trap states? If this is the case, How they affect the performance? The effect is more significant in Fig. S3b than in Fig. 2b, how the synthesis conditions affects this behavior and what are the implications in the device performance?
- Fig. 5d shows and increase of performance of the solar cells during the first days after preparation. What is the origin of this increase? What are the storage conditions (atmosphere, light, humidity...)? I guess efficiencies reported in Fig. 5c are the ones obtained few days after the fabrication, If this is the case authors have to indicate it.

Reviewer #3 (Remarks to the Author):

Wang et al. present a novel synthetic approach allowing to easily obtain all-inorganic PbS quantum dots (QDs) in scale-up amount quantity at room temperature. The authors demonstrated promising potential of these QDs for solar cell and photodetector applications. From a manufacturing standpoint, the results of the work may be considered interesting, further extending the compositional diversity of Pb-based crystals (e.g. methylammonium lead triiodide) for inexpensive solution-processing deposition techniques. However, from scientific point of view, readers will not find much in this paper. The work is limited to the synthesis of PbS QDs only, with poor control over the size of the QDs, their shape and size distribution (>17%) as well as packing ordering. Moreover, nothing new was presented in terms of surface chemistry and improved performance in comprising to already published data by the Sargent group that were numerously cited in the current manuscript.

Reviewer #1 (Remarks to the Author):

The manuscript “Room-temperature direct synthesis of semi-conductive PbS nanocrystal inks for optoelectronic applications” is an interesting piece of work for the effective upscaling of PbS quantum dot (QD) fabrication with a significant reduction of the production cost and no significant reduction of performance (even an increase). The authors can synthesize PbS QDs at room temperature and prepare inks able to produce photodetectors and solar cells with performances at level of the state-of-the-art. Taking into account these points I consider that the manuscript could be interesting for the readers of Nature Communications. Nevertheless, I would suggest the revision of some parts:

- Future commercial applications of PbS QDs could be restricted by the use of Pb. The synthesis route reported here is appealing but the use of DMF, that also awake important concerns, likely will not help in the commercialization. Did the authors used another solvents? This discussion is mandatory if the final scope is the commercialization.

Response: Firstly we'd like to thank the reviewer for the very positive comments on the significance of our work. We agree with the reviewer that the DMF is not an optimal solvent for commercialization. Thus we chose a more environmentally benign solvent γ -butyrolactone (GBL) for the synthesis of PbS-I NCs. As shown in Figure S6, we have successfully synthesized PbS-I NCs using non-toxic GBL as the reaction solvent [*Clinical Toxicology*, **2012**, 50, 458.]. In addition, as shown in Fig.S17, devices based on GBL synthesized PbS-I NCs show comparably high performance as well.

Corrective actions:

On page 5, line 17, we have added the following sentence:

“It is worth noting that DMF is not an optimal solvent for commercialization. Thus we chose a more environmentally benign solvent γ -butyrolactone (GBL) for the synthesis of PbS-I NCs. As shown in Figure S6, we have successfully synthesized PbS-I NCs using non-toxic GBL as the reaction solvent. In addition,

as shown in Fig.S17, devices based on GBL synthesized PbS-I NCs show comparably high performance as well.”

In the supporting information, we have added the following figures:

Figure S6. Absorption and PL spectrum of PbS-I NCs synthesized with GBL as the reaction solvent.

Figure S17. *J-V* curve of solar cell based on PbS-I NCs synthesized with GBL as the reaction solvent.

- PL measurements (Fig. 2b y S3b) shows a clear queue or shoulder at long wavelengths, is it originated by trap states? If this is the case, How they affect the performance? The effect is more significant in Fig. S3b than in Fig. 2b, how the synthesis conditions affects this behavior and what are the implications in the device performance?

Response: We are sorry for the confusion in the measurement of PL spectra. After checking on the PL measurement set-up, It turns out that those shoulders are due to the lack of correction file. We then measured again with the correction file (1000nm~1600nm). As shown in Fig. 2b and Fig.S3, there are no more shoulders with the employ of correction file. Furthermore, as the reviewer suggested, we have investigated the effect of synthesis conditions on the corresponding device performance. Solar cells based on PbS QDs synthesised with different Pb/S ratio and precursor concentrations were fabricated. As shown in Fig.S13, the devices using QDs with Pb/S (4/1) ratio and high concentration (0.8M) show the best device performance.

Corrective actions:

We have changed Fig.2a, Fig.2b and Fig.S3. In addition, we have added the device performance data in Fig.S13.

Figure 2. Characterization of direct-synthesized PbS-I NCs. Absorption (a) and PL spectrum (b) of synthesized PbS-I NCs with different Pb/S Precursor ratio.

Figure S3. Absorption (a) and PL spectrum (b) of PbS-I NCs synthesized from different precursor concentration. Pb/S precursor ratio was fixed at 4:1.

Figure S13. Average device performance based on PbS-I NCs synthesized with different precursor concentration (a) and lead/sulphur ratio (b).

At page 10, line 19, we added the following sentences:

“In addition, PbS-I NCs synthesized from concentrated solution, within the solubility limit of the precursors, show increased performance, as shown in Fig. S13.”

- Fig. 5d shows an increase of performance of the solar cells during the first days after preparation. What is the origin of this increase? What are the storage conditions (atmosphere, light, humidity...)? I guess efficiencies reported in Fig. 5c are the ones obtained few days after the fabrication, if this is the case authors have to indicate it.

Response: This initial improvement in performance over short-term air exposure is widely observed. [*Nat. Mater.*, **2017**, 16, 258–263., *Nat. Mater.*, **2014**, 13, 796–801.] This improvement is mainly due to the oxidation of under-charged Pb in PbS NCs array films, which would passivate part of the trap states and lead to a better band alignment. In nowadays, air exposure is a routine procedure in PbS QDs device fabrication. [*RSC Adv.*, **2018**, 8, 15149–15157., *Adv.Mater.*, **2015**, 27, 4481–4486.] We are sorry for the omission of the storage conditions. Now, we have included it in the experimental section. In addition, we have mentioned as well that the highest efficiency is obtained after 5 days' air exposure.

Corrective actions:

At page 9, line 20, we have added “After 5 days' air storage, ...”

In experimental section, we have added, “Devices are stored in ambient air (humidity ~30%) for the air stability test.”

Reviewer #3 (Remarks to the Author):

Wang et al. present a novel synthetic approach allowing to easily obtain all-inorganic PbS quantum dots (QDs) in scale-up amount quantity at room temperature. The authors demonstrated promising potential of these QDs for solar cell and photodetector applications. From a manufacturing standpoint, the results of the work may be considered interesting, further extending the compositional diversity of Pb-based crystals (e.g. methylammonium lead triiodide) for inexpensive solution-processing deposition techniques. However, from scientific point of view, readers will not find much in this paper. The work is limited to the synthesis of PbS QDs only, with poor control over the size of the QDs, their shape and size distribution (>17%) as well as packing ordering. Moreover, nothing new was presented in terms of surface chemistry and improved performance in comprising to already published data by the Sargent group that were numerous cited in the current manuscript.

Response: We thank the reviewer for pointing out our contribution in reducing the manufacturing cost, which is in agreement with the opinion of another reviewer.

We have to emphasize that, for solar cells, three topics are primary: efficiency, stability and cost. For the former two topics, lots of work have been done on PbS quantum dot solar cells, while few effort has been paid to reduce the cost of PbS QDSCs. Recently, one paper pointed out that the high cost of PbS QDSCs is mainly due to the synthesis and ligand exchange process (*Energy Environ. Sci.*, **2018**, 11, 2295.). The source of Pb and S are relatively cheap and abundant, which means there is no fundamental reason for the high cost of current PbS QDSCs; and the development of new low-cost synthetic methods is critically important for the commercial relevance of QD photovoltaics. As the reviewer pointed out here, in this work we didn't focus on QD surface chemistry or device performance since numerous works have been done on these issues. In this study, we only devoted to address the issue of cost reduction, which is equally important and very few work has been done on it.

Our convenient synthetic method can not only dramatically reduce the cost of PbS NC inks, but also provide relatively high performance devices, which is quite decent for a new synthetic method. We firmly believe further investigation would improve the NC quality and achieve higher-performance devices. Furthermore, up to now, only a few groups over the world can fabricate high performance PbS quantum dots solar cells, which is also mainly due to the tedious and subtle processes for PbS QDs synthesis and ligand exchange. With our robust and facile method, it will be much easier to fabricate high performance PbS quantum dot solar cells, which, we believe, would accelerate the performance improvement of these devices.

Note that, as the reviewer suggested, we have tried to extend our method to synthesize other semi-conducting metal chalcogenide NCs, such as Ag₂S and CdS QDs, which have been often reported in photovoltaic devices [*Nano. Lett.* **2011**, 11, 3998., *ChemNanoMat* **2018**, 4, 1223.]. Now, we have successfully synthesized Ag₂S and CdS NCs using our method, which indicates that our synthetic method shows generality in obtaining semi-conducting metal chalcogenide NCs for photovoltaic application. Unfortunately, because lead based perovskite will decompose in DMF, we are not able to synthesize perovskite NCs using our method.

Corrective actions:

At page 6, line 14; we have added a paragraph about the synthesis of Ag₂S and CdS NCs:

“Furthermore, considering Ag₂S and CdS are also commonly used materials in photovoltaic devices, we further explored the synthesis of halide capped Ag₂S and CdS NCs. With some modifications (See Experimental Section), Accanthite phase Ag₂S NCs and Hawleyite phase CdS NCs were successfully synthesized (Fig. S7 and S8) with I⁻ and Cl⁻ ions as the capping ligands, respectively. These results indicate the versatility of this synthetic method and its potential to become a general synthetic routine for obtaining other metal chalcogenide NCs inks.”

Figure S7. Directly synthesized Ag_2S -I NCs. (a) Absorption spectrum of Ag_2S -I NCs. (b) XRD of Ag_2S -I NCs. (c) TEM image of Ag_2S -I NCs.

Figure S8. Directly synthesized CdS -CI NCs. (a) Absorption and PL spectrum of CdS -CI NCs. (b) XRD of CdS -CI NCs. (c) TEM image of CdS -CI NCs.

Reviewers' comments:

Reviewer #1 (Remarks to the Author):

From my point of view, authors have revised properly the manuscript taking into account the reviewers comments. I suggest to publish the revised version.

Reviewer #3 (Remarks to the Author):

Wang et al. present a novel synthetic approach allowing to easily obtain all-inorganic PbS quantum dots (QDs) in scale-up amount quantity at room temperature. The authors demonstrated the promising potential of these QDs for solar cell and photodetector applications. From a manufacturing standpoint, the results of the work may be considered interesting, further extending the compositional diversity of Pb-based crystals (e.g. methylammonium lead triiodide) for inexpensive solution-processing deposition techniques. However, from scientific point of view, readers will not find much in this paper. The work is limited to the synthesis of PbS QDs only, with poor control over the size of the QDs, shape and size distribution (>17%) as well as packing ordering. Moreover, nothing new was presented in terms of surface chemistry and improved performance in comprising to already published data by the Sargent group that were numerously cited in the current manuscript.

Reviewer #4 (Remarks to the Author):

In this manuscript, Ma et al. report a novel method to prepare PbS QD inks at room temperature. Using metal halide and N, N-Diphenyl thiourea (DPhTA) as precursors to synthesis larger amount in-situ halide passivated PbS QDs is innovative to a certain extent, but there are less explanation and discussion about this reaction in the manuscript. For mass synthesis of PbS QDs, J. B. Zhang reported a method to synthesis larger amount chloride passivated PbS and PbSe QDs in 2014 (Zhang et al. ACS Nano, 2014, 8, 614-622). Although TMS was used as S source in that work, narrow size distributions (6~7%) QDs were obtained which is smaller than this work. Except the new synthesis method, I can not find much new scientific results in this manuscript. Thus, I think I cannot recommend it to be published in Nature Communications in the current form.

1. More focus on the discussion and explanation of the reaction between metal halide and DPhTA may improve the scientific values of this manuscript.

2. How about the purity of the obtained QDs power? The precursors ratio for Pb and DPhTA is very large in the reaction, but the authors only add some anti-solvent in to the reaction solution and without any subsequent purification treatment. Have all by-products and raw materials been removed from the QDs power?

3. In GBL reaction system, 4 mmol PbI₂, 4 mmol MAI and 1 mmol DPhTA, 9 mL GBL were used. When toluene was added into the reaction solution, the unreacted PbI₂ and MAI have a large potential to form perovskite (MAPbI₃). Has the author considered this question? I very much doubt the reliability of this strategy.

We thank the other two reviewers for their positive attitude towards publication of our revised manuscript. The point-to-point responses of the reviewer #4's questions are listed below.

Reviewer #4 (Remarks to the Author):

In this manuscript, Ma et al. report a novel method to prepare PbS QD inks at room temperature. Using metal halide and N, N-Diphenyl thiourea (DPhTA) as precursors to synthesis larger amount in-situ halide passivated PbS QDs is innovative to a certain extent, but there are less explanation and discussion about this reaction in the manuscript. For mass synthesis of PbS QDs, J. B. Zhang reported a method to synthesis larger amount chloride passivated PbS and PbSe QDs in 2014 (Zhang et al. ACS Nano, 2014, 8, 614-622). Although TMS was used as S source in that work, narrow size distributions (6~7%) QDs were obtained which is smaller than this work. Except the new synthesis method, I cannot find much new scientific results in this manuscript. Thus, I think I cannot recommend it to be published in Nature Communications in the current form.

Response: We thank the reviewer for pointing out the innovation of our new synthesis method. As suggested by the reviewer, we have presented detailed reaction mechanism in the revised manuscript.

We want to mention that the innovation of our method is not limited to the synthesis scalability. As we have demonstrated in our manuscript, so far, PbS NCs have to be first synthesized with long alkyl chain ligands and then be exchanged with shorter ones (two-steps) to enable charge transport. However, the initial synthesis of insulated PbS NCs show no necessity and the ligand exchange process is tedious and costly. Our synthesis can use green solvent like γ -butyrolactone and cheaper precursors, providing PbS NCs ink directly applied to optoelectronic devices. Our method totally avoids the ligand exchange step, which can not only largely reduce the material cost but also eliminate the possibility of trap introduction during the ligand exchange step. Even though our current PbS NCs ink has relatively broad size distribution, a decent PCE of 10.1 % has been achieved, which also indicates the huge potential of our method. More efforts will also be devoted to improving size distribution in the near future. Collectively, we firmly believe our new strategy contributes to reduce fabrication cost and possess potential to further improve device performance of NCs solar cells.

1. More focus on the discussion and explanation of the reaction between metal halide and DPhTA may improve the scientific values of this manuscript.

Response: Thanks for this valuable suggestion. We have now added detailed discussion and explanation of the reaction between metal halide and DPhTA as shown below.

Corrective actions:

In Page 5, line 7, we have added the following discussion about the reaction mechanism:

“After butyl amine (BA) was injected into the aforementioned solution, the solution turned from light yellow to black immediately, indicating the formation of PbS NCs. Since PbI₂ and DPhTA cannot react in the absence of BA, we speculate that the ⁻SH converted from DPhTA under alkaline condition works as the real sulfur precursor (Fig. S1) (*Chem. Mater.* **2013**, 25, 1233., *J. Am. Chem. Soc.* **1956**, 78, 5560.). We also tried to replace DPhTA with TMS-S, which can react with PbI₂ directly. The reaction can indeed be triggered without BA and, however, result in large size PbS precipitates (Fig. S2). If BA was introduced, the obtained PbS NCs show almost similar size and dispersibility with our PbI₂+DPhTA+BA recipe, which indicates BA also function as temporal surface ligands to control the nucleation and growth in addition to promote the release of ⁻SH from DPhTA. The detailed discussion of reaction mechanism was presented in the Supporting Information.”

In the supporting information, we have added a section of “Proposed Reaction Process”:

“Before discussing our direct synthesis of PbS NCs ink, we briefly review the mechanism of the typical synthesis for oleate capped PbS NCs through hot-injection reaction. For conventional hot injection synthesis method, lead oxide, lead acetate or lead halides are used as lead precursors; bis(trimethylsilane sulphide) (TMS-S) or oleylamine (OLA)-S are used as sulphur precursor; oleic acid (OA) or OLA are used as ligands (*Adv. Mater.* **2003**, 15, 1844., *ACS Nano* **2014**, 8, 614., *ACS Nano* **2014**, 8, 6363., *Nat. Mater.* **2014**, 13, 796.). The reaction can be summarized as follow:

In the recipe (1), the highly reactive TMS-S can easily react with metal salts (e.g., halides, acetates, metal alkyls), forming metal chalcogenide NCs and TMS-X (X = halogen, acetate, alkyl) (*Phil. Trans. R. Soc. A* **2010**, 368, 1455).

In the recipe (2), S powder is converted to thioamide, which is further converted to H₂S. The generated H₂S can react with metal precursors to form metal sulfide NCs. And it has also been confirmed thioacetamide dissolved in OLA can work as sulfur precursor, which can react with Pb precursor to produce PbS NCs even at room temperature (*J. Am. Chem. Soc.* **2011**, 133, 5036.). In addition, in this reaction OLA also work as surface ligand to control NCs nucleation and growth. But the OLA can only bind on NCs surface weakly, which needs to be exchanged with OA.

In our direct synthesis method,

since PbI_2 and DPhTA cannot react in the absence of BA, it should be reasonable to propose that S^- converted from DPhTA under alkaline condition works as the real sulfur precursor (*Chem. Mater.* **2013**, 25, 1233., *J. Am. Chem. Soc.* **1956**, 78, 5560), analogous to reaction (2). Then we can propose the detailed process of our reaction as follow:

Figure S1. Scheme diagram for the reaction between DPhTA, BA and PbI_2 .

In order to confirm if BA also function as ligands in our reaction system, we use TMS-S, instead of DPhTA, as sulphur precursor. As shown in Fig. S2, PbS NCs can be obtained with TMS-S as well. This reaction can also be triggered even without BA, but it can only result in black PbS precipitates with large size and serious aggregations, which indicates BA also functions as surface ligand to control the nucleation and growth in our direct ink synthesis (reaction 3).

Since BA binds on PbS NCs surface weakly and possesses low boiling point (78°C). It will be easily removed during purification and spin-coating step. The special property of BA can ensure low residual of organic compound in the final coated film, which is beneficial for charge transport. It is also the same reason BA was used as solvent to dissolve PbS ink for solar cells fabrication. (*Nat. Mater.* **2017**, 16, 258., *Nat. Nanotechnol.* **2018**, 13, 456.)

Figure S2. PbS-I NCs direct-synthesized with TMS-S as sulfur precursor. (a) Absorption spectrum of PbS NCs synthesized with BA. (b) TEM image of PbS NCs (w/ and w/o BA). (c) Photos of reaction solution with and without butyl amine.

2. How about the purity of the obtained QDs powder? The precursors ratio for Pb and DPhTA is very large in the reaction, but the authors only add some anti-solvent in to the reaction solution and without any subsequent purification treatment. Have all by-products and raw materials been removed from the QDs powder?

Response: We understand the reviewer's concern, since the Pb precursor was 4 times excess than S in our reaction.

a) We have to admit it was hard to fully purify the NCs. Since further purification will lead to aggregation of the NCs, which is the same situation as in the solution-phase ligand exchange process.

b) Furthermore, it is believed that the small amount residual PbI_2 will not cause negative effect on device performance, since the PbI_2 would help to passivate PbS NCs. In the recently reported high-efficiency PbS QDs solar cells, the NCs were exactly passivated by PbI_2 . (*Nat. Mater.* **2017**, 16, 258., *Nat. Nanotechnol.* **2018**, 13, 456.)

c) As we have mentioned, the NCs were also purified only once in conventional solution-phase ligand exchange process, which gave the device performance more than 12%. (*Nat. Nanotechnol.* **2018**, 13, 456.) In that case, 0.5mmol PbI_2 and 0.1mmol PbBr_2 are used as ligands for 50mg PbS-OA QDs. For our direct synthesis method, ~ 0.45 mmol PbI_2 is used as both lead precursor and ligands for 50mg PbS-I NCs. The excess of lead halide in our method is much less than solution-phase ligand-exchange method. Furthermore, there is no obvious peak from crystalline PbI_2 phase in the resultant PbS as shown in the XRD pattern (Fig. 2e). which indicates the residual PbI_2 is really limited. Collectively, the amount of residual PbI_2 is very little, which would actually have positive passivation effect.

We also performed XPS survey spectra (Fig.S11) and FTIR spectrum (Fig.S12) to check the amount of organic by-products. As shown in Fig. S11, PbS-I NCs films contain Pb, S, I, C and small amount O and N. The main components are Pb, S and I. The FTIR spectrum (Fig.S12) only shows very small amount of butylamine, and no obvious peaks for other organic components can be clearly identified, which indicates the amount of organic by-products is too low to be detected.

Corrective actions:

In the supporting information, we have added following figures:

Figure S11. XPS survey spectrum of PbS-I NCs film.

Figure S12. FTIR spectrum of PbS-I NCs film.

3. In GBL reaction system, 4 mmol PbI_2 , 4 mmol MAI and 1 mmol DPhTA, 9 mL GBL were used. When toluene was added into the reaction solution, the unreacted PbI_2 and MAI have a large potential to form perovskite (MAPbI_3). Has the author considered this question? I very much doubt the reliability of this strategy.

Response: Due to the limited solubility of PbI_2 in GBL, we have to add MAI to promote its dissolution. We agree with the reviewer that if we add toluene into the GBL solution

containing PbI_2 and MAI, black materials, which should be perovskite (MAPbI_3), will precipitate out (Figure R1 middle). Considering there exists a large amount of BA in our system, the perovskite structure could not survive, due to the good solubility of PbI_2 and MAI in BA. As confirmed in Figure R1 right, the black perovskite totally decomposes after adding BA. Furthermore, we also measured the XRD of obtained PbS NCs with GBL as the reaction solvent. As shown in Fig.S8, except cubic phase PbS, no obvious diffraction peaks of perovskite can be observed, which further confirmed the absence of perovskite in our PbS NCs powder.

Figure R1. Photos of PbI_2 /MAI/GBL solution with addition of toluene and butylamine. The ratios of the chemicals are same as that in our reaction.

Corrective actions:

In the supporting information, we have added the following figure:

Figure S9. XRD spectrum of PbS-I NCs synthesized with GBL as the reaction solvent.

REVIEWERS' COMMENTS:

Reviewer #4 (Remarks to the Author):

The authors have revised the manuscript carefully according to the comments. Now I would like to recommend it to be published in Nature Communications.